# EFFICIENT LOW-RANK DIFFUSION MODEL TRAINING FOR TEXT-TO-IMAGE GENERATION

## ABSTRACT

Recent advancements in text-to-image generation models have witnessed the success of large-scale diffusion-based generative models. However, exerting control over these models, particularly for structure-conditioned text-to-image generation, remains an open challenge. One straightforward way to achieve control is via fine-tuning, often coming at the cost of efficiency. In this work, we address this challenge by introducing ELR-Diffusion (Efficient Low-rank Diffusion), a method tailored for efficient structure-conditioned image generation. Our innovative approach leverages the low-rank decomposition of model weights, leading to a dramatic reduction in memory cost and model parameters — by up to 58%, at the same time performing comparably to larger models trained with expansive datasets and more computational resources. At the heart of ELR-Diffusion lies a two-stage training scheme that resorts to the low-rank decomposition and knowledge distillation strategy. To provide a robust assessment of our model, we undertake a thorough comparative analysis in the controllable text-to-image generation domain. We employ a diverse array of evaluation metrics with various conditions, including edge maps, segmentation maps, and image quality measures, offering a holistic view of the model's capabilities. We believe that ELR-Diffusion has the potential to serve as an efficient foundation model for diverse user applications that demand accurate comprehension of inputs containing multiple conditional information.

## 1 INTRODUCTION

In the realm of text-to-image (T2I) generation, diffusion models exhibit exceptional performance in transforming textual descriptions into visually accurate images. Such models exhibit extraordinary potential across a plethora of applications, spanning from content creation (Rombach et al., 2022; Saharia et al., 2022b; Nichol et al., 2021; Ramesh et al., 2021b; Yu et al., 2022; Chang et al., 2023), image editing (Balaji et al., 2022; Kawar et al., 2023; Couairon et al., 2022; Zhang et al., 2023; Valevski et al., 2022; Nichol et al., 2021; Hertz et al., 2022; Brooks et al., 2023; Mokady et al., 2023), and also fashion design. As the field strives for more control over image generation, enabling more targeted, stable, and accurate visual outputs, several models like T2I-Adapter (Mou et al., 2023), Composer (Huang et al., 2023), and Uni-ControlNet (Zhao et al., 2023) have emerged which aim to enhance control over the image generation process. Despite their prowess, ensuing challenges arise, particularly concerning memory usage, computational requirements, and a thirst for extensive datasets (Saharia et al., 2022a; Rombach et al., 2022;

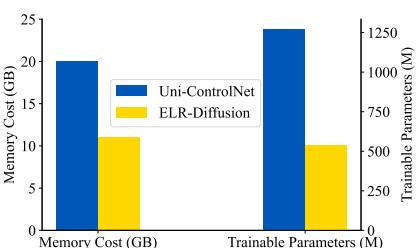

Figure 1: The comparison of computation costs between our method and Uni-ControlNet (Zhao et al., 2023). Our method significantly reduces the overall memory cost by 45% and a number of trainable parameters by 58%, in the meanwhile largely retaining the performance.

Ramesh et al., 2021a). Controllable text-to-image generation models also often come at significant computational costs, facing challenges such as linear growth in costs and size when dealing with different conditions.

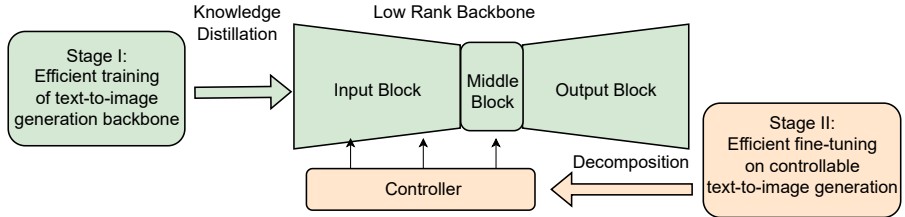

Figure 2: The overview pipeline of our method. Our method improves the efficiency of controllable text-to-image generation from two aspects. At Stage I, we propose an efficient pre-training method for the standard text-to-image generation via knowledge distillation. For Stage II, we propose to resort to low-rank and Kronecker decomposition to reduce the tunable parameter space.

To address these challenges, we present ELR-Diffusion, an innovative method tailored for the U-Net architecture integral to diffusion-based T2I models. This method adeptly harnesses low-rank structures to streamline the training of T2I diffusion models. We adopt a two-stage training approach: First, we train a lightweight T2I diffusion model based on a standard U-Net architecture. This is bolstered by the inclusion of knowledge distillation during the training phase. Next, we fine-tune the model to enhance its capabilities for controlled T2I tasks. Such two-stage strategy leads to substantial reductions in the resources needed for fine-tuning and also curtails the total number of model parameters. Figure 1 presents a comparative analysis between our approach and Uni-ControlNet (Zhao et al., 2023), particularly focusing on memory consumption and trainable parameters.

Upon being trained on a sizable dataset encompassing 5 million text-image pairs, ELR-Diffusion stands out by delivering performance metrics comparable to its counterparts—even those trained on far larger datasets (e.g., 10 million text-image pairs from the LAION dataset)—as illustrated in Figure 2. Central to our strategy is including a distinctive multi-scale knowledge distillation process. This involves guiding a novice or 'Student' diffusion model using feature maps that draw upon the wisdom of a more seasoned 'Teacher' model. For tasks centered around controlled T2I generation, we've deployed a shared, Stable Diffusion encoder, reducing the number of trainable parameters and memory cost via synergizing the prowess of low-rank formulations with the intricacies of the Kronecker product. This aids in converting input conditions into conditional tokens, subsequently channeled through a cross-attention mechanism. A pivotal insight from our study lies in the mathematical congruence between the low-rank training processes across both training phases, unveiling the symmetries in low-rank training trajectories across both phases.

Our primary contributions are summarized as follows:

- We propose ELR-Diffusion, a novel text-to-image generation model for efficient controllable image generation tasks that substantially reduces training memory overhead and model parameters concurrently.

- We propose a two-stage training scheme, at Stage I, we design a lightweight U-Net structure via the low-rank decomposition and involve knowledge distillation in the training process. At Stage II, we bring down the parameter and memory cost through low-rank decomposition over the controller.

- ELR-Diffusion shows on-par performance with the Uni-ControlNet baseline with overall 58% trainable parameters and 45% training memory. Quantitative results validate the effectiveness of our method on comprehensive metrics for controllable text-to-image generation tasks.

## 2 METHOD

An overview of our proposed two-stage pipeline is shown in Figure 2. In Stage I, we train the U-Net of a text-to-image model with a low-rank schema. Specifically, we employ matrix factorization techniques that decompose high-dimensional matrices into smaller matrices, capturing essential features with reduced computational overhead. This process is augmented through knowledge

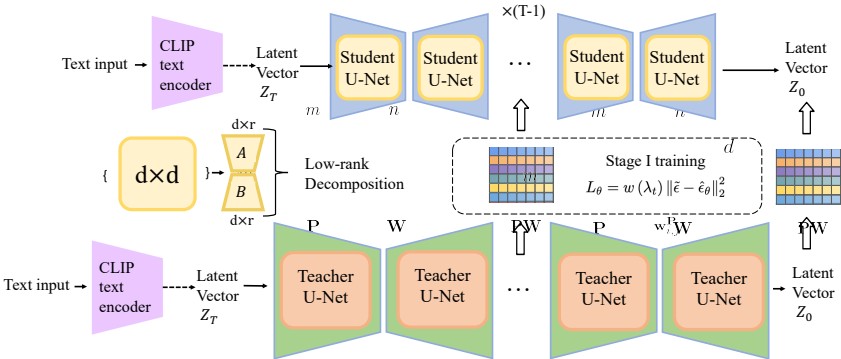

Figure 3: Overview of the Stage-1 training: Training a low-rank U-Net using knowledge distillation from a teacher model (green) to the student model (blue). This process involves initializing the student U-Net with a decomposition into low-rank matrices and minimizing the loss between the predicted noise representations from the student and teacher.

distillation, visually represented in green on Figure 2. We then conduct efficient fine-tuning at Stage II (shown in yellow part on Figure 2), where we employ low-rank decomposition and Kronecker decomposition to streamline the parameter space.

## 2.1 BACKGROUND ON LOW-RANK TRAINING

**Background on Training in Low-dimensional Space**   Let $\theta^D = \left[ \theta_0{}^D \ldots \theta_m{}^D \right]$ be a set of $m$ $D$-dimensional parameters that parameterize the U-Net within the Stable Diffusion. Instead of optimizing the noise prediction loss in the original parameter space $\left( \theta^D \right)$, we are motivated to train the model in the lower-dimensional space $\left( \theta^d \right)$ (Aghajanyan et al., 2020). Our overall pipeline is trying to train the controllable text-to-image diffusion model in such a lower-dimension space to improve the overall efficiency.

## 2.2 STAGE I: TRAINING A LOW-RANK U-NET

**Low-rank Text-to-image Diffusion Model**   To establish a foundational understanding of our model, it's crucial to first comprehend the role of U-Nets in the diffusion process. In diffusion models, there exists an input language prompt $y$ that is processed by a text encoder $\tau_\theta$. This encoder projects $y$ to an intermediate representation $\tau_\theta(y) \in \mathbb{R}^{M \times d_\tau}$, where $M$ is denotes the token length, and $d_\tau$ denotes the dimension of the embedding space . This representation is subsequently mapped to the intermediate layers of the U-Net through a cross-attention layer given by

$$\text{Attention}(Q, K, V) = \text{softmax}\left( \frac{QK^T}{\sqrt{d}} \right) V, \tag{1}$$

with $Q = \mathbf{W}_Q \varphi_i\left(z_t\right), \quad K = \mathbf{W}_K \tau_\theta(y), \quad V = \mathbf{W}_V \tau_\theta(y)$. In this context, $\varphi_i\left(z_t\right) \in \mathbb{R}^{N \times d_\epsilon}$ is an intermediate representation of the U-Net. The terms $\mathbf{W}_V \in \mathbb{R}^{d \times d_\epsilon}, \mathbf{W}_Q \in \mathbb{R}^{d \times d_\tau}, \mathbf{W}_K \in \mathbb{R}^{d \times d_\tau}$ represent learnable projection matrices.

Shifting focus to the diffusion process, during the t-timestep, we can represent:

$$K = \mathbf{W}_K \tau_\theta(y) = AB\tau_\theta(y), \quad V = \mathbf{W}_V \tau_\theta(y) = AB\tau_\theta(y), \tag{2}$$

where $A$ and $B$ are decomposed low-rank matrices from the cross-attnetion matrices, $d_\tau$ and $d_\epsilon$ denote the dimension for the text encoder and noise space respectively. Conventionally, the diffusion model is trained via minimizing $\mathcal{L}_\theta = \left\| \epsilon - \epsilon_\theta \right\|_2^2$, where $\epsilon$ is the groundtruth noise and $\epsilon_\theta$ is the predicted noise from the model.

To fully exploit the prior knowledge from the pre-trained teacher model while exploiting less data and training a lightweight diffusion model, we propose a new two-stage training schema. The first one is the initialization strategy to inherit the knowledge from the teacher model. Another is the knowledge distillation strategy. The overall pipeline is shown in Figure 3.

### 2.2.1 INITIALIZATION

Directly initializing the student U-Net is not feasible due to the inconsistent matrix dimension across the Student and teacher U-Net. We circumvent this by decomposing U-Net into two low-rank matrices, namely $A$ and $B$ for the reconstruction. We adopt an additional transformation to adapt the teacher's U-Net weights to the Student, which leverages the Singular Value Decomposition (SVD) built upon the teacher U-Net. The initialization process can be expressed as:

1. Compute the SVD of the teacher U-Net: Starting with the teacher U-Net parameterized by $\theta_0$, we compute its SVD as $\theta_0 = U \Sigma V^T$.

2. Extract Low-Rank Components: to achieve a low-rank approximation, we extract the first $k$ columns of $U$, the first $k$ rows and columns of $\Sigma$, and the first $k$ rows of $V^T$. This results in matrices $U_k$, $\Sigma_k$, and $V_k^T$ as follows:

   $U_k = $ first $k$ columns of $U$, $\ \Sigma_k = $ first $k$ rows & columns of $\Sigma$, $\ V_k^T = $ first $k$ rows of $V^T$

3. We then initialize the student U-Net with $U_k \Sigma_k$ and $V_k^T$ that encapsulate essential information from the teacher U-Net but in a lower-rank format.

We observe in practice that such initialization effectively retains the prior knowledge inherited from Teacher U-Net while enabling the student U-Net to be represented in a compact form thus computationally more efficient for later training.

### 2.2.2 LOSS FUNCTION

We propose to train our Student U-Net with knowledge distillation (Meng et al., 2023) to mimic the behavior of a teacher U-Net. This involves minimizing the loss between the student's predicted noise representations and those of the teacher. To be specific, our training objective can be expressed as:

$$\mathcal{L}_\theta = w(\lambda_t) \left\| \tilde{\epsilon} - \hat{\epsilon}_\theta \right\|_2^2, \tag{3}$$

where $\tilde{\epsilon}$ denotes the predicted noise in the latent space of Stable Diffusion from the teacher model, $\hat{\epsilon}_\theta$ is the corresponding predicted noise from the student model, parameterized by $\theta$, and $w(\lambda_t)$ is a weighting function that may vary with the time step $t$. Such an objective encourages the model to minimize the squared Euclidean distance between the teacher and Student's predictions thus providing informative guidance to the Student. We also tried combining the loss with the text-to-image Diffusion loss but using our training objective works better.

## 2.3 DISCUSSION: CONNECTION BETWEEN LORA AND OUR EFFICIENT TRAINING PARADIGM

In this section, we discuss the connections between LoRA (Low-Rank Adaptation) fine-tuning (Hu et al., 2021) and our proposed efficient training method, centering on the low-rank structure of the model.

### 2.3.1 LORA FINE-TUNING V.S. ELR-DIFFUSION

LoRA optimizes a model by constraining the rank of the difference between the fine-tuned parameters, $\theta$, and the initial parameters $\theta_0$. The objective can be expressed as:

$$rank(\theta - \theta_0) = k, \tag{4}$$

where $k$ is the pre-determined rank constraint. This method transforms the optimization problem into a constrained lower-dimensional subspace.

In contrast to LoRA, our method introduces a low-rank U-Net architecture by designating the weights at initialization as low-rank so that:

$$rank(\theta_0) = k, \tag{5}$$

where $\theta_0$ denotes low-rank learnable parameters. Instead of introducing extra parameters for dimensionality reduction, we aim to facilitate efficient training by reducing the dimensionality of parameters.

### 2.3.2 EQUIVALENCE PROOF

We note that our proposed efficient training is intrinsically similar to LoRA in that they optimize identical target functions. In specific:

**LoRA Paradigm:** Given an input $x$, LoRA computes the output $h$ in such a way that:

$$h = (\theta_0)^\top x + (\Delta\theta)^\top x = (\theta_0 + \Delta\theta)^\top x = (\theta_0 + (\mathbf{T} - \mathbf{I}) \cdot \theta_0)^\top x, \tag{6}$$

where $\Delta\theta$ is the change of parameters from fine-tuning, and $\mathbf{T}$ is a general transformation matrix.

**Our Paradigm:** For our method, the output $h$ is obtained with a low-rank initialized parameter $\theta_0^d$ so that:

$$h = (\theta_0^d)^\top x = (\mathcal{T}(\theta_0))^\top x = (\theta_0 + (\mathbf{T} - \mathbf{I}) \cdot \theta_0)^\top x. \tag{7}$$

From the equations above, it is shown that both methods transform the optimization objective into a lower-dimensional space. LoRA achieves this via fine-tuning separate parameters, whereas our method takes advantage of a low-rank U-Net structure from the initialization stage.

### 2.4 STAGE II: FINE-TUNING FOR CONTROLLABLE TEXT-TO-IMAGE GENERATION

To achieve flexible control over the generated images and harness the potential of the foundational model developed in Stage I, we delve into the second stage of our approach. This stage aims to integrate enhanced control mechanisms without ballooning the computational overhead.

**Local Control Adapter and Condition Injection** Inspired by ControlNet, we fix the weights of a Stable Diffusion (SD) model (Rombach et al., 2022) and use it as a backbone. We leverage the network architecture and weights of the encoder and middle block from the U-Net backbone, denoting them as $F'$ and $M'$, respectively. This architecture and the weights inherit the low-rank format from our Stage I training. We modify the input of the decoder to gradually integrate control information into the main backbone. Unlike ControlNet, which directly applies conditions to input noises, we adopt a multi-scale condition injection strategy that extracts features at different resolutions and uses them for condition injection referring to the implementation of Feature Denormalization (FDN) in Zhao et al. (2023), expressed as:

$$\text{FDN}(Z, c) = \text{norm}(Z) \cdot (1 + \Phi(\text{zero}(h_r(c)))) + \Phi(\text{zero}(h_r(c))), \tag{8}$$

where $Z$ denotes noise features, $c$ denotes the input conditional features, $\Phi$ denotes learnable convolutional layers, and $\text{zero}$ denotes zero convolutional layer. The zero convolutional layer contains weights initialized to zero. This ensures that during the initial stages of training, the model relies more on the knowledge from the backbone part, gradually adjusting these weights as training progresses. The use of such layers aids in preserving the architecture's original behavior while introducing structure-conditioned inputs.

**Parameter Reduction through Low-Rank Decomposition** Building on the low-rank methods used during Stage I training, we hypothesize that the update weights of the copied encoder, denoted as $\Delta\boldsymbol{W}$, can be adapted in a low-rank subspace via Kronecker decomposition and low-rank decompostion to further reduce parameter cost. This leads to:

$$\Delta\boldsymbol{W} = \sum_{i=1}^{n} \boldsymbol{A_i} \otimes \left(u_i v_i^\top\right), \tag{9}$$

with $u_i \in \mathbb{R}^{\frac{k}{n} \times r}$ and $v_i \in \mathbb{R}^{r \times \frac{d}{n}}$, where $r$ is the rank of the matrix which is a small number, $\boldsymbol{A_i}$ are the decomposed learnable matrices, and $\otimes$ is the Kronecker product operation. The low-rank decomposition ensures a consistent low-rank representation strategy. This approach substantially saves trainable parameters, allowing efficient fine-tuning of the copied encoder.

# 3 EXPERIMENTS

In this section, we embark on evaluating our ELR-Diffusion method across two stages, focusing primarily on the task of controllable text-to-image (T2I) generation. By controllable generation, we are referring to the generation of images based on structured input conditions, specifically the sketch maps, edge maps, and the segmentation maps. We begin by detailing the datasets( 3.1) we have utilized, followed by the diverse set of evaluation metrics employed to quantitatively assess our model's performance ( 3.2). Our experimental setup ( 3.3) provides insight into the specific models and techniques we have incorporated, after which we present a comparative study of our approach against the widely recognized Uni-ControlNet in the controllable text-to-image generation domain ( 3.4). We then offer a synthesis of our findings through tables that showcase efficiency and performance measures ( 3.5), culminating with a qualitative analysis contrasting our results with Uni-ControlNet across various structural input conditions.

## 3.1 DATASETS

In alignment with our goal of controlled T2I generation, we employed the LAION-6+ (Schuhmann et al., 2022) dataset. We first curated a subset consists of 5,082,236 examples by deduplicating, omitting images with copyright and NSFW issues, and filtering based on resolution. Given the controlled generation tasks we target, we collect training data based on additional input conditions such as sketch maps, edge maps, and the segmentation maps, The feature extraction process for these maps was based on the methodology presented in (Zhang & Agrawala, 2023).

## 3.2 EVALUATION METRICS

Our model's performance is assessed using various metrics tailored to different aspects of the generated output:

- **Normalized Mean Square Error (NMSE):** We use the Normalized Mean Squared Error to quantify the difference between the predicted and actual edge maps. For the generated images, we extract the edge mapsr (Canny, 1986) and then measure the MSE between the predicted map and the actual map. A lower value signifies that the predicted maps closely resemble the actual maps.

- **Intersection over Union (IoU)** (Rezatofighi et al., 2019): Intersection over Union measures the overlap between the predicted and ground truth segmentation maps. A higher IoU score demonstrates better segmentation accuracy.

- **FID:** We use FID (Heusel et al., 2017) to measure the realism and variation of the generated images. A lower FID indicate superior quality and diversity of the output images.

- **CLIP Score** (Hessel et al., 2021; Radford et al., 2021)[1]: We use CLIP Score to measure the semantic similarity between the generated images and the input text prompts, as well as the sketch maps. The strength of the CLIP Score lies in its ability to embed both visual and textual information into a shared semantic space, making it a potent metric to assess the consistency between visual outputs (like sketch maps and generated images) and textual descriptions.

## 3.3 EXPERIMENTAL SETUP

We employed the Stable Diffusion 2.1 [2] model in conjunction with xFormers (Lefaudeux et al., 2022) and FlashAttention (Dao et al., 2022) using the implementation available in HuggingFace Diffusers. [3] Our approach emphasizes efficiency improvements through the decomposition of model weights, particularly the U-Net architecture. For the computational setup, our experiments were mainly conducted on AWS EC2 instances, specifically on P3 instances containing 64 NVIDIA V100 GPUs for fine-tuning. In Stage I, we used the standard training scheme of Stable Diffusion Rombach et al. (2022) without the classifier-free guidance (Ho & Salimans, 2022). In Stage II, as a same setup with

---

[1]`https://github.com/jmhessel/clipscore`
[2]`https://huggingface.co/stabilityai/stable-diffusion-2-1`
[3]`https://huggingface.co/docs/diffusers/index`

Table 1: Comparing U-Net models: Original, decomposed, with and without Knowledge Distillation. ELR-Diffusion-Stage I showcases a promising balance between performance and efficiency. Note that compared with Stable Diffusion, ELR-Diffusion-Stage I is only trained on 5 million data. ELR-Diffusion-Stage I beats Decomposed U-Net w/o Distillation interms of FID and CLIP Score, suggesting the effectiveness of our distillation strategy in training the decomposed U-Net.

| Methods | FID↓ | CLIP Score↑ | # Parameters ↓ |
|---|---|---|---|
| Stable Diffusion | 27.7 | 0.824 | 1290M |
| Standard U-Net w/o Distill. | 66.7 | 0.670 | 1290M |
| Decomposed U-Net w/o Distill. | 84.3 | 0.610 | 790M |
| ELR-Diffusion-Stage I | 45.0 | 0.768 | 790M |

Table 2: Efficiency and controllability comparison: Our method vs. Uni-ControlNet for text-to-image generation. Notably, ELR-Diffusion-Stage II exhibits significant reductions in memory cost, trainable parameters, and training time, while maintaining competitive performance metrics across various tasks.

| | Metric | Uni-ControlNet | Ours-Stage II |
|---|---|---|---|
| **Efficiency** | Memory Cost ↓ | 20GB | 14GB |
| | # Params. ↓ | 1271M | 750M |
| | Training Time ↓ | 5.72s/it | 2.15s/it |
| **Performance** | Sketch Maps (CLIP Score ↑) | 0.49 | 0.45 |
| | Edge Maps (NMSE ↓) | 0.60 | 0.60 |
| | Segmentation Maps (IoU ↑) | 0.70 | 0.74 |
| | Image Quality (FID ↓) | 27.7 | 27.5 |

Uni-ControlNet, we trained our model with 1 epoch. We adopted the AdamW optimizer (Kingma & Ba, 2014) with the learning rate of $10^5$. In all of our experiments, we resized the input images and extracted conditional images to resolutions of $512 \times 512$.

## 3.4 BASELINES

We compared our method against Uni-ControlNet, a widely used baseline in the text-to-image generation task. Uni-ControlNet is designed for unified controllable text-to-image generation, balancing the trade-off between model complexity and expressive power. Our method further explores potential efficiency improvement through model decomposition and low-rank optimization while maintaining comparable performance.

## 3.5 RESULTS

Table 1 illustrates the comparison between different variations of our method in Stage I, including original U-Net, decomposed low-rank U-Net, and their respective performance with and without knowledge distillation. It is observed that the decomposed low-rank U-Net models demonstrate efficiency gains, with a reduction in the total number of parameters to 790M, although at the cost of some fidelity in metrics such as FID and CLIP Score. Employing distillation helps to mitigate some of these performance reductions.

From Table 2, it is evident that our method offers significant efficiency gains over Uni-ControlNet, with a 30% reduction in memory cost and a decrease in trainable parameters from 1271M to 750M, with the original full number of parameters in Stage II being 1478M. At the same time, the training time cost per iteration reduces from 5.72s to 2.15s. Our approach maintains the original full parameter count while optimizing the model's efficiency. Table 2 highlights our method's performance in terms of the metrics for different input conditions in comparison to Uni-ControlNet. While showing a

Table 3: Performance and resource metrics comparison of ELR-Diffusion with the baseline Uni-ControlNet. The ELR-Diffusion approach with distillation shows a significant reduction in resource consumption while providing competitive image quality and outperforming in controllability metrics, especially in segmentation maps. The $\Delta$ column shows the improvement of ELR-Diffusion (w/o distillation) compared with no distillation.

| | Metrics | Uni-ControlNet | ELR-Diffusion | | $\Delta$ |
| | | | w/o Distill. | w/ Distill. | |
|---|---|---|---|---|---|
| **Efficiency** | Memory Cost ↓ | 20GB | **11GB** | **11GB** | **0** |
| | # Params. ↓ | 1271M | **536M** | **536M** | **0** |
| **Image Quality** | FID ↓ | 27.7 | 84.0 | 43.7 | **- 40.3** |
| | CLIP Score ↑ | 0.82 | 0.61 | 0.77 | **+0.16** |
| **Controllability** | Sketch Maps (CLIP Score)↑ | 0.49 | 0.40 | 0.46 | **+0.06** |
| | Edge Maps (NMSE ) ↓ | 0.60 | 0.54 | 0.57 | **+0.03** |
| | Segmentation Maps (IoU) ↑ | 0.70 | 0.40 | 0.74 | **+0.34** |

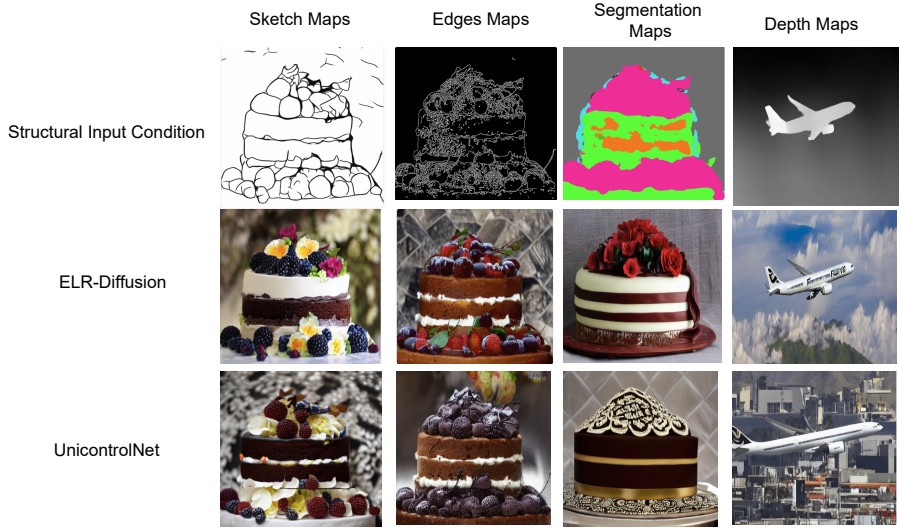

Figure 4: Qualitative comparison of ELR-Diffusion finetuned for controllable text-to-image generation task with Uni-ControlNet as baseline. ELR-Diffusion shows comparable performance with the Uni-ControlNet.

slight decrease in F1 score for edge maps, our method outperforms Uni-ControlNet in Normalized MSE, IoU, and Inception Score. This demonstrates our approach's capability to enhance performance across various aspects of text-to-image generation, including edge maps, depth maps, segmentation maps, and image quality.

Table 3 illustrates the comparison between our method and the baseline training end-to-end. It is observed that the decomposed low-rank U-Net models demonstrate efficiency gains, with a reduction in the total number of parameters to 790M, although at the cost of some fidelity in metrics such as FID and CLIP Score. Employing distillation helps to mitigate some of these performance reductions.

These collective results affirm our method's capability to not only enhance efficiency but also improve or maintain performance across various aspects of text-to-image generation.

### 3.6 QUALITATIVE RESULTS

We show the qualitative comparison of Stage II in Figure 4. As can be seen, ELR-Diffusion can achieve competitive results with the baseline Uni-ControlNet under three different structural input conditions: segmentation map, canny edges, and sketch edges.

## 4 RELATED WORK

ELR-Diffusion is an instance of efficient training in the vision-and-language domain. Here, we overview modeling efforts in the subset of efficient training towards reducing parameters and memory cost as well as knowledg distillation strategies.

**Efficient Training** Prior work has proposed efficient training methodologies both for pre-training and fine-tuning. These methods have established their efficacy across an array of language and vision tasks. One of these explored strategies is Prompt Tuning (Lester et al., 2021), where trainable prompt tokens are appended to pre-trained models (Schick & Schütze, 2020; Ju et al., 2021; Jia et al., 2022). These tokens can be added exclusively to input embeddings or to all intermediate layers (Li & Liang, 2021), allowing for nuanced model control and performance optimization. Low-Rank Adaptation (LoRA) (Hu et al., 2021) is another innovative approach that introduces trainable rank decomposition matrices for the parameters of each layer (Hu et al., 2021). LoRA has exhibited promising fine-tuning ability on large generative models, indicating its potential for broader application. Furthermore, the use of Adapters inserts lightweight adaptation modules into each layer of a pre-trained transformer (Houlsby et al., 2019; Rücklé et al., 2021). This method has been successfully extended across various setups (Zhang et al., 2021; Gao et al., 2021; Mou et al., 2023), demonstrating its adaptability and practicality. Other approaches including post-training model compression (Fang et al., 2023) facilitate the transition from a fully optimized model to a compressed version – either sparse (Frantar & Alistarh, 2023), quantized (Li et al., 2023; Gu et al., 2022), or both. This methodology was particularly helpful for parameter quantization (Dettmers et al., 2023). Different from these methodologies, our work puts forth a new unified strategy that aims to enhance the efficient training of text-to-image diffusion models through the leverage of low-rank structure. Our proposed method integrates principles from these established techniques to offer a fresh perspective on training efficiency, adding to the rich tapestry of existing solutions in this rapidly evolving field.

**Knowledge Distillation for Vision-and-Language Models** Knowledge distillation (Gou et al., 2021), as detailed in prior research, offers a promising approach for enhancing the performance of a more streamlined "student" model by transferring knowledge from a more complex "teacher" model (Hinton et al., 2015; Sanh et al., 2019; Hu et al.; Gu et al., 2021; Li et al., 2021). The crux of this methodology lies in aligning the predictions of the student model with those of the teacher model. While a significant portion of existing knowledge distillation techniques leans towards employing pre-trained teacher models (Tolstikhin et al., 2021), there has been a growing interest in online distillation methodologies (Wang & Jordan, 2021). In online distillation (Guo et al., 2020), multiple models are trained simultaneously, with their ensemble serving as the teacher. Our approach is reminiscent of online self-distillation, where a temporal and resolution ensemble of the student model operates as the teacher. This concept finds parallels in other domains, having been examined in semi-supervised learning (Peters et al., 2017), label noise learning (Bengio et al., 2010), and quite recently in contrastive learning (Chen et al., 2020). Our work on distillation for pre-trained text-to-image generative diffusion models distinguishes our method from these preceding works. Salimans & Ho (2022); Meng et al. (2023) propose distillation strategies for diffusion models but they aim at improving inference speed. Our work instead aims to distill the intricate knowledge of teacher models into the student counterparts, ensuring both the improvements over training efficiency and quality retention.

## 5 CONCLUSION

This work introduces an efficient optimization approach for diffusion-based text-to-image generation. Our experimental results demonstrate a substantial reduction in memory cost and trainable param-

eters without compromising inference time or performance. Our study offers an exciting avenue toward more resource-efficient models, paving the way for text-to-image generation applications on constrained hardware or for large-scale deployments. The potential applications of our method are vast, suggesting its importance in the continually evolving domain of image synthesis. Future work may explore more sophisticated decomposition techniques and their impact on different architectures, furthering the pursuit of an optimal balance between model efficiency, complexity, and expressive power.

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
