# OpenReview forum: "Efficient Low-Rank Diffusion Model Training for Text-to-Image Generation"
_ICLR.cc/2024/Conference — ICLR 2024 Conference Withdrawn Submission_

### Official Review · Reviewer_svyE · 2023-10-28

**Soundness:** 2 fair
**Presentation:** 1 poor
**Contribution:** 1 poor
**Rating:** 1
**Confidence:** 5

**Summary:**

The paper proposes a parameter-efficient training scheme for controllable T2I generation. The essence of the method is to convert memory cost for fine-tuning into time cost by introducing a knowledge distillation step. Specifically, the paper introduces a two-stage approach. In stage I, the diffusion model with attention layers whose linear matrices could be decomposed into two low-rank matrices is trained by knowledge distillation. In stage II, the diffusion model is fine-tuned for controllable T2I generation via  Feature Denormalization (FDN).

**Strengths:**

- The proposed method is parameter-efficient for the fine-tuning stage: it can reduce the number of tuned parameters to half of the original diffusion model.
- The proposed initialization method using truncated SVD is intriguing as it can preserve a portion of the performance of the teacher model.

**Weaknesses:**

- **The paper contains exactly the same sentences for one section as another paper.**
- Technical novelty is not significant. Ideas of knowledge distillation, and low-rank constrained fine-tuning have been widely used not only for the diffusion model but also for various foundation models. The paper just combines them to convert the memory cost into time cost.
- The efficiency of the proposed method is doubtable.
- Comparison with LoRA is missing, although LoRA also offers a parameter-efficient method to get a desired model with the same memory requirements (11GB GPU VRAM).
- There are multiple issues in writing: repeated sentences, notations without description, over-stated sentences, and wrong descriptions.

**Questions:**

- The usage of the section 2.1 is not clear. The reviewer could not find a clear connection to the proposed method.

-  In equation (3), does the low-rank decomposition for linear transformations always hold? Empirically, we know that $\Delta W$ after fine-tuning of attention layers shows low-rankness (namely LoRA), but the reviewer wonders if it could be a general assumption for any linear matrices.

- The description of diffusion model training in section 2.2 is too simplified. The sentence does not clearly show the objective function and the meaning of the trained diffusion model. In parallel, the equation (4) is limited to offer enough information about the proposed method. What should be given as input to $\tilde{\epsilon}$ and $\hat{\epsilon}_\theta$? Is it noisy images or noisy latent representations? Does the method use classifier-free guidance (CFG)? It would be helpful to express the equations precisely.

- It is doubtful that 10GB memory reduction is worth even with significant performance degradation. Actually, one can simply conduct LoRA fine-tuning for Stable-Diffusion on 11GB of GPU RAM (https://huggingface.co/docs/diffusers/main/en/training/lora) without additional knowledge distillation process which also raises a question about the efficiency of the proposed method.

- Comparison with LoRA fine-tuning is required to analyze the efficiency of the proposed method. Due to the knowledge distillation stage, the proposed method may lose its capability and the important aspect for the efficiency is how much.

- In Table 2, only run-time for each iteration is computed to compare the efficiency of the methods, but it is a trivial result since the number of learnable parameters is reduced. To fairly compare the efficiency of the proposed method, the entire time to get a task-specific model should be computed because it is the time that users really consume to get their own model. Especially, the time for stages I and II should be summed since the proposed method requires additional knowledge distillation processes that play a key role in reducing memory cost by allowing low-rank initialization before fine-tuning. It would be helpful to analyze the efficiency of the proposed method if the paper reports the training time for the entire procedure.

- The performance of the proposed method is different in Tables 2 and 3, while the performance of the baseline and metrics are the same. There is no mention about the difference in the experimental setup.

- In table 3, the performance difference $\Delta$ is computed for different baselines. For efficiency, the baseline is Uni-ControlNet, but for the qualities, the baseline is the proposed method without knowledge distillation which is closer to the ablation study. This would be confusing since this seems to say the performance is better than Uni-ControlNet, even if the actual result is not.

- Based on table 3, the quality of the generated image by the proposed method is worse than the baseline. However, in figure 4, all examples show that the proposed method is better than the baseline method which is not aligned with the quantitative comparison. It would be better to display more examples for the qualitative comparison in the main paper or appendix to precisely see where the proposed method is located.

- The writing could be improved with precise descriptions. For example,
	- In the second contribution in the introduction section, the paper does not state that the low-rank decomposition is also leveraged for Stage 1.
	- In section 2.2, the meaning of "domain-specific encoder $\tau_\theta$" contains ambiguity. Does the domain mean modalities such as text and image, or does it mean categories of given language prompts such as English and French? Also, no description for $M, d_\tau, z_t$, the superscript $i$ for linear transformations, and $\tau$ in equation (3).
	- Redundant expressions in section 2.4: "This architecture and the weights inherit the low-rank format..." and following "These weights are represented in a low-rank format..."
	- No description for $A_i$ matrix which is introduced by the Kronecker decomposition (equation 10). Thus, it is not clear what the proposed method is intended to learn (i.e. what are learnable parameters) exactly.
	- In section 3.5, the third sentence in the second paragraph says that "our approach maintains the original full parameter count". However, the proposed method actually reduces the number of parameters by the truncated SVD in stage I.
	- Repeated sentences in the caption of Table 3.

**Details Of Ethics Concerns:**

- Section 2.1 is the same as Aghajanyan et al [1]. Especially, the paper repeats the wrong description of the original sentences as is:  $\theta^D = [ \theta_0,...,\theta_m ]$  should be a set of $m$ parameters on $D$-dimensional space according to the following contexts, but the original paper and the submitted paper wrote it as "a set of $D$ parameters". The last sentence seems to be rephrased but is wrongly converted: the projection operation as  $\mathcal{P}: \mathbb{R}^D \rightarrow \mathbb{R}^d$, which is actually $\mathcal{P}: \mathbb{R}^d \rightarrow \mathbb{R}^D$. The reviewer thinks this is plagiarism and raises a flag for ethics review.

[1] Intrinsic Dimensionality Explains the Effectiveness of Language Model Fine-Tuning, arXiv:2012.13255

---

> ### Author Response · Authors · 2023-11-20
>
> Thanks for your hard work, careful reviews and useful comments! We would like to clarify several major misunderstandings about this work. Please see our responses below:
>
>
>
>
> *Background section 2.1*:
> In our Background section 2.1, we refer to the definition and formulation of re-parameterization from Aghajanyan et al [1] to lay the groundwork for our proposed method. The philosophy behind our work and the referred work is the same, this is the reason for including the background section 2.1. In Stage I, we employ low-rank decomposition to break down the U-Net model, enabling training in a reduced-dimensional space. In contrast to Aghajanyan et al [1], where the Fastfood transform is utilized for optimization in a lower dimension (d) and then projected back to a higher dimension (D), our approach differs significantly. Specifically, in Stage I, our model undergoes direct decomposition and training in this lower-dimensional space without any subsequent projection back. Stage II involves fine-tuning a smaller subset of parameters compared to the original full model. This fine-tuning occurs initially through Kronecker decomposition followed by low-rank decomposition, and these modifications are directly added to the original parameters in D dimensions. Hence, we introduce the operator \( \mathcal{P} \) for projecting from D to d dimensions, distinguishing our notation from that used in Aghajanyan et al [1]. To avoid confusion, we will revise our phrasing regarding 'a set of D parameters' to clearly indicate a set of parameters in D-dimensional space. Additionally, we will clarify the initial sentence "Let \( \theta_D = [\theta_0, \theta_1, ..., \theta_m] \) be a set of D parameters that parameterize some model \( f(·, \theta) \)" for better understanding.
>
>
> *Technical novelty*
> Our contribution lies in the novel combination of low-rank decomposition with knowledge distillation in an end-to-end approach, which to our knowledge, is the first of its kind. This integration not only reduces memory costs but also preserves the effectiveness of the diffusion models. Besides, our application in the context of text-conditioned diffusion models, especially the controllable text-to-image generation tasks presents unique challenges. We make the Kronecker decomposition first and then the low-rank decomposition, with the decomposed weights shared across different conditions. We will further elucidate this aspect in our revised manuscript, highlighting the unique challenges and solutions in this context.
>
> *Comparison with LoRA*
> We will add baseline comparisons with LoRA in the revision. We also want to clarify the 10GB memory reduction happens on the Uni-ControlNet model rather than the vanilla Stable Diffusion, which contains more modules and parameters.
>
> *Writing Issues*: We will be thoroughly reviewed and corrected to improve the clarity and accuracy of the manuscript in the revision.
>
>
> *Low-rank decomposition for linear transformations*:
> We want to clarify that equation (3) applies the decomposition for attention-layers, specifically for \(W_k\) and \(W_v\) matrices. Furthermore, as shown in the LoRA paper [1], they freeze the MLP modules from the consideration of simplicity and parameter-efficiency and only apply it on attention-matrices. Apply the decomposition on more weights can bring additional costs.
>
> [1] LoRA: Low-Rank Adaptation of Large Language Models
>
>
> *Time to get a task-specific model*:
> We train our model and baselines for the same number of iterations and that’s why we report the training time per iteration. Table II compares the training time for the individual Stage II, where the standard Stable Diffusion without distillation can also be used as the backbone. We propose Stage I to distillate a lightweight model that can serve as an off-the-shelf pretrained model and it’s not necessary to perform Stage I and II each time so it is meaningless to sum them together for comparison.
>
> *Difference in the experimental setup*:
> Table 2 compares the cost only for Stage II while Table 3 shows the overall cost.
>
> *Qualitative Comparison*: We will include additional experiments and analyses that compare our method with other baselines in additional datasets. We will add more qualitative results. We will also display more examples for the qualitative comparison in the revision.

---

### Official Review · Reviewer_aWRN · 2023-10-30

**Soundness:** 2 fair
**Presentation:** 3 good
**Contribution:** 2 fair
**Rating:** 3
**Confidence:** 4

**Summary:**

The paper proposed a diffusion model compression and efficient fine-tuning framework. It first compresses the diffusion model via model distillation and then introduces a low-rank efficient fine-tuning approach with the low-rank assumption of the student model. The paper verifies the proposed distillation can reduce the model size, memory footprint, and improve the fine-tuning efficiency without hurt the model performance significantly.

**Strengths:**

1. The paper is well written and easy to follow.
2. The evaluation verifies the contributions, including model distillation can reduce the diffusion model size, low-rank fine-tuning can reduce the memory requirement, etc.

**Weaknesses:**

1. The novelty of this paper is somewhat trivial. The proposed two stage approach is a simple combination of model distillation and low rank fine-tuning. The paper uses the vanilla model distillation and low rank fine-tuning without much innovation.

2. The paper did not explain how to improve the controllability of the generation model by the proposed approach. There is no evidence to show how the control is improved. The visualization results in Figure 4 is not sufficient to justify the capability improvement of controlling the generation process.

3. The proposed approach distills the model first and fine-tunes it for downstream tasks in the second step. The distillation does not involve the downstream tasks information which could hurt the performance. Would it make more sense to fine-tune the model first and distill it then?

4. The proposed fine-tuning assumes the matrix is low-rank. There are no ablation studies that verify such an assumption. In general, the distilled model should be denser than the teacher model. Would the low-rank assumption hurt the performance if the matrix is not low-rank?

5. The evaluation is not sufficient to justify the contribution. For example, there is no evaluation between LORA and the proposed approach.

6. Some theoretical analysis is confusing. For example, in eq 8, \theta_0^d should be expressed as a SVD transformation of \theta_0 instead of a linear transformation of \theta_0. Can authors clarify how eq 8 is derived?

**Questions:**

Extra questions besides the questions in weakness 3, 4, and 6

7. The paper assumes the distillation is model architecture distillation. Would the step distillation help here?

---

> ### Author Response · Authors · 2023-11-20
>
> We are grateful for the detailed feedback from the reviewer. Please find our responses below.
>
> *Novelty of Approach*:
> Our contribution lies in the novel combination of low-rank decomposition with knowledge distillation in an end-to-end approach, which to our knowledge, is the first of its kind. This integration not only reduces memory costs but also preserves the effectiveness of the diffusion models. Besides, our application in the context of text-conditioned diffusion models, especially the controllable text-to-image generation tasks presents unique challenges. We will further elucidate this aspect in our revised manuscript, highlighting the unique challenges and solutions in this context.
>
>
> *Improvement in Controllability*: We want to clarify that the target of this work is not to  improve the controllability of the generation model by the proposed approach but to improve the overall efficiency of training such a model.
>
>
> *Model Distillation and Fine-Tuning Sequence*: We distill the model first followed by fine-tuning because the downstream task requires a pre-trained Stable Diffusion backbone, and we perform distillation first to obtain a lightweight backbone so that we can perform fine-tuning based on it. However, your suggestion to reverse this sequence is intriguing. We will explore this alternative approach and include the findings in our revised manuscript.
>
>
> *Low-Rank Assumption*: We appreciate your observation on the low-rank assumption. To address this, we will conduct and include ablation studies in our revised paper to validate this assumption and examine its impact on performance.
>
>
> *Comparison with LoRA*: We would like to clarify that we didn’t assume the matrix is low-rank across both our two stages. In the first stage, rather than assume the U-Net is low rank, we perform low-rank decomposition over the model. In the second stage, we perform low-rank finetuning which can work on dense matrices as well. In the revised manuscript, we will include these comparisons to demonstrate the relative strengths and weaknesses of our method.
>
>
> *Theoretical Clarifications*: Thank you for pointing out the need for clarification in our theoretical analysis, particularly concerning Equation 8. We would like to clarify that in Equation 8, \theta_0^d  is indeed intended to represent a transformation of \theta_0, but not strictly a linear one. Instead, \theta_0^d is derived through a more general transformation operator, which we denote as 'T'. This operator 'T' encompasses a broader range of transformations, which includes but is not limited to Singular Value Decomposition (SVD).

---

### Official Review · Reviewer_mXK5 · 2023-11-01

**Soundness:** 2 fair
**Presentation:** 2 fair
**Contribution:** 1 poor
**Rating:** 3
**Confidence:** 5

**Summary:**

This work proposes a neural network compression method for fine-tuning text-conditioned diffusion models. They distill a larger U-Net backbone in LDM to a smaller student U-Net using low-rank decomposition. They compare their results to Uni-ControlNet and show decreased computation costs compared to the baseline.

**Strengths:**

1. The recent generative models require large-scale training and increasing their efficiency for training and fine-tuning is an interesting and important problem.

2. The paper is well-written and is easy to follow.

**Weaknesses:**

1. Although the task is image generation, there are only a few qualitative samples. Also, the quantitative metrics only include FID and CLIP score. The authors could also include KID and Inception Score.

2. The performance loss in FID and CLIP Score is large. The loss of image quality is also visually visible in the qualitative results.

3. The combination of low-rank decomposition for knowledge distillation is not novel and has been used in previous works.

4. There are a lot of ambiguities in details such as the teacher and student architectures, the amount of fine-tuning, the hyperparameters, etc. These make the work not reproducible.

5. Although there have been many network compression methods, none of these has been compared against in the results.

[a] Li, Tianhong, et al. "Few sample knowledge distillation for efficient network compression." CVPR 2020.
[b] Dai, Cheng, et al. "A tucker decomposition based knowledge distillation for intelligent edge applications." Applied Soft Computing 101 (2021): 107051.
[c] Gu, Yuchao, et al. "Mix-of-Show: Decentralized Low-Rank Adaptation for Multi-Concept Customization of Diffusion Models." arXiv preprint arXiv:2305.18292 (2023).
[d] Yao, Xufeng, et al. "Distill Vision Transformers to CNNs via Low-Rank Representation Approximation." (2022).
[e] Rui, Xiangyu, et al. "Unsupervised Pansharpening via Low-rank Diffusion Model." arXiv preprint arXiv:2305.10925 (2023).
[f] Ryu, Simo, Seunghyun Seo, and Jaejun Yoo. "Efficient Storage of Fine-Tuned Models via Low-Rank Approximation of Weight Residuals." arXiv preprint arXiv:2305.18425 (2023).

**Questions:**

1. Since the architecture used in ELR has less number of parameters compared to Uni-ControlNet, it requires fewer steps and data for fine-tuning. Has this been considered in the experiments?

---

> ### Author Response · Authors · 2023-11-20
>
> We thank the reviewer for the constructive suggestions. Your suggestions will be incorporated in our next revision. Please see our clarifications and responses below:
>
> *Qualitative and Quantitative Results*: In the revised manuscript, we will include additional qualitative examples to better illustrate the capabilities of our method. Furthermore, we will expand our quantitative analysis to include KID and Inception Score, alongside FID and CLIP Score, to provide a more comprehensive evaluation.
>
> *Novelty of Approach*: Our contribution lies in the novel combination of low-rank decomposition with knowledge distillation in an end-to-end approach, which to our knowledge, is the first of its kind. This integration not only reduces memory costs but also preserves the effectiveness of the diffusion models. Besides, our application in the context of text-conditioned diffusion models, especially the controllable text-to-image generation tasks presents unique challenges. [a][b][f] propose knowledge distillation and low-rank decomposition methods mainly for convolutional neural networks, intelligent edge applications, and language models, while ours propose specific methods for diffusion-based text-to-image generation models. [c] proposes to use LoRA to compose multiple customized concepts, while ours propose an end-to-end efficient training strategy that can deal with different input conditions such as segmentation maps and depth maps. [e] proposes a low-rank matrix factorization technique for panchromatic (PAN) image, where the task is also completely different from ours.
>
> [a] Li, Tianhong, et al. "Few sample knowledge distillation for efficient network compression." CVPR 2020.
>
> [b] Dai, Cheng, et al. "A tucker decomposition based knowledge distillation for intelligent edge applications." Applied Soft Computing 101 (2021): 107051.
>
> [c] Gu, Yuchao, et al. "Mix-of-Show: Decentralized Low-Rank Adaptation for Multi-Concept Customization of Diffusion Models." arXiv preprint arXiv:2305.18292 (2023).
>
> [d] Yao, Xufeng, et al. "Distill Vision Transformers to CNNs via Low-Rank Representation Approximation." (2022).
>
> [e] Rui, Xiangyu, et al. "Unsupervised Pansharpening via Low-rank Diffusion Model." arXiv preprint arXiv:2305.10925 (2023).
>
> [f] Ryu, Simo, Seunghyun Seo, and Jaejun Yoo. "Efficient Storage of Fine-Tuned Models via Low-Rank Approximation of Weight Residuals." arXiv preprint arXiv:2305.18425 (2023).
>
> *Experimental Details*: We will provide a more detailed description of both the teacher and student architectures, the extent of fine-tuning, and the specific hyperparameters used in the revision.
>
> *Responding to the Question*:
> Regarding the fewer steps and data required for fine-tuning due to the reduced number of parameters in ELR compared to Uni-ControlNet, this aspect was considered in our experiments. As can be seen from our experimental results in Table 2, the training time required is less than half of the Uni-ControlNet and we train EIR-diffusion the same number of iterations with Uni-ControlNet.
> As for data, we only require 5 million text-image pairs, while Uni-ControlNet requires 10 million. Ours saves 50%, demonstrating the data efficiency of EIR-Diffusion.

---

> > ### Comment · Reviewer_mXK5 · 2023-11-23
> >
> > I would like to thank the authors for the rebuttal. I share the same concerns as the other reviewers. The authors promise further comparisons and results in the revision, however judging the work without these additional results is not possible. Furthermore, I still think the novelty of applying the low-rank decomposition combined with knowledge distillation to image generation is limited for ICLR.

---

### Official Review · Reviewer_HH76 · 2023-11-05

**Soundness:** 3 good
**Presentation:** 2 fair
**Contribution:** 2 fair
**Rating:** 3
**Confidence:** 4

**Summary:**

This paper introduces an approach for compressing diffusion models designed explicitly for refining text-conditioned diffusion models. The paper discusses several related works in efficient training, including Prompt Tuning, Low-Rank Adaptation (LoRA), and Adapters.  Compared to these existing methods, the paper offers an approach for efficient training and fine-tuning in text-to-image generation. It leverages the low-rank decomposition of model weights, leading to a reduction in memory cost and model parameters. The paper presents a two-stage training scheme, where a smaller U-Net structure is trained with low-rank schema in the first stage, and fine-tuning is conducted in the second stage. The technique involves distilling a more cumbersome U-Net backbone in the Latent Diffusion Model (LDM) into a more compact with comparable or less effective student U-Net by applying low-rank decomposition. The performance of this method is compared with that of Uni-ControlNet, demonstrating a reduction in computational costs relative to the baseline in unconditioned and conditioned settings. This suggests that the proposed method offers a more efficient alternative for fine-tuning text-conditioned diffusion models. For evaluation, the authors reported Normalized Mean Square Error (NMSE), Intersection over Union (IoU), Fréchet Inception Distance (FID), and CLIP Score in data from the LAION-6+ dataset to assess the performance of the proposed ELR-Diffusion method compared to the Uni-ControlNet model in terms of edge maps, segmentation maps, and image quality.

**Strengths:**

The paper claims that ELR-Diffusion achieves comparable performance to larger models trained with expansive datasets and more computational resources while reducing the memory cost by 45% and the number of trainable parameters by 58% (specific experiments).

The literature review was interesting to follow and read.

**Weaknesses:**

Since the low-rank decomposition has been proposed in previous works and would not be considered a new approach, further and more extensive evaluations in qualitative and quantitative results sections would help the readers understand how effective this method is compared to other baselines. I also would like to see experiments on how this compression can affect image generation in other domains and datasets. It would also be interesting to see its effect on different resolutions as well.

The experiment settings need to be in more detail for the sake of reproducibility,

**Questions:**

Have you considered the effect of different schedulers on how the compression would be effective?
What are the results on other datasets?
Have you noticed any exciting phenomenon, negative or positive, in the qualitative results? I am curious if compression could lower the diversity or maybe decrease the performance of particular objects. E.g., a comparison between small vs large things.

---

> ### Author Response · Authors · 2023-11-20
>
> We thank the Reviewer for the constructive feedbacks. We value your specific suggestions, which we have carefully considered in our responses below and we will include them in our next revision.
>
> *Novelty of Low-Rank Decomposition*: Our contribution lies in the novel combination of low-rank decomposition with knowledge distillation in an end-to-end approach, which to our knowledge, is the first of its kind. This integration not only reduces memory costs but also preserves the effectiveness of the diffusion models. Besides, our application in the context of text-conditioned diffusion models, especially the controllable text-to-image generation tasks presents unique challenges.
>
>
> *Extensive Evaluations*: We will include additional experiments and analyses that compare our method with other baselines in diverse domains, datasets, and resolutions. This will provide a clearer understanding of the effectiveness and versatility of our method.
>
>
> *Detailed Experiment Settings*: In our revised manuscript, we will provide more comprehensive details of our experimental setup, ensuring that other researchers
> can replicate our work accurately.
>
> *Additional Ablations*: we used DDIM scheduler in the inference as the standard scheduler for accelerating the text-to-image generation. In the revised manuscript, we will include results and observations related to the use of various schedulers in our method. We are currently conducting experiments on other datasets and will include these results in our revised submission. These additional experiments will provide a broader view of how our method performs across different task setups.
>
> *Effects on compression*: we want to clarify that the qualitative results contain the task of controllable text to image generation, where the generation is regularized by the additional input condition. In this context, our method produces results that are comparable in terms of diversity and quality. We have observed some interesting phenomena in this regard, both positive and negative. For instance, we noted a decrease in the performance of image quality at higher compression levels when increasing the number of decomposed matrices to 16. We will add more qualitative results in the revision.

---

> > ### Comment · Reviewer_HH76 · 2023-11-23
> > **Response to rebuttal**
> >
> > Thank you to the authors for their answers. I am not convinced that the shortcomings can be addressed unless a considerable amount of changes happen.